# Association of daytime napping with incidence of chronic kidney disease and end-stage kidney disease: A prospective observational study

Qinjun Li[1,2◉], Ying Shan[1,3◉], Jingchi Liao[4◉], Ling Wang[1,2], Yanling Wei[1,3], Liang Dai[1,3], Sen Kan[1,2], Jianqing Shi[4,5], Xiaoyan Huang[1,3]*, Guoyuan Lu[1]*

1 Department of Nephrology, The First Affiliated Hospital of Soochow University, Suzhou, Jiangsu, China, 2 Department of Nephrology, The Affiliated Xuzhou Municipal Hospital of Xuzhou Medical University, Xuzhou, Jiangsu, China, 3 Renal Division, Department of Clinical Research Academy, Peking University Shenzhen Hospital, Shenzhen, Guangdong, China, 4 Department of Statistics and Data Science, Southern University of Science and Technology, Shenzhen, Guangdong, China, 5 Department of Statistics and Data Science, National Center for Applied Mathematics, Shenzhen, Guangdong, China

◉ These authors contributed equally to this work.

* luguoyuan@suda.edu.cn (GL); huangxiaoyan@pku.org.cn (XH)

**Data Availability Statement:** Yes - all data are fully available without restriction; The UK Biobank resource is available to bona fide researchers for health-related research in the public interest. All

## Abstract

### Background and aims

Few studies have examined the relationship between daytime napping and risk of kidney diseases. We aimed to investigate the association of daytime napping with the incidence of chronic kidney disease (CKD) and end-stage kidney disease (ESKD). We also examined whether sleep duration modified the association of nap with CKD or ESKD.

### Methods

We recruited 460,571 European middle- to older-aged adults without prior CKD or ESKD between March 13, 2006, and October 1, 2010, in the UK Biobank. Sleep behavior data were obtained through questionnaires administered during recruitment. The analysis of the relationship between napping and the occurrence of CKD and ESKD utilized Cox proportional hazards regression models. The modification role of sleep duration on the effect of nap on CKD and ESKD was also examined.

### Results

After a mean follow-up of 11.1 (standard deviation 2.2) years, we observed 28,330 incident CKD cases and 927 ESKD cases. The daytime napping was associated with incident CKD (P for trend = .004). After fully adjusted, when compared with participants who did not take nap, those in sometimes and usually nap groups had higher risk of CKD. Nevertheless, the available evidence did not support a link between daytime napping and ESKD (P for trend = .06). Simultaneously, there was insufficient evidence suggesting that sleeping duration modified the association of daytime napping with incident CKD or ESKD.

researchers who wish to access the research resource must register with UK Biobank by completing the registration form in the Access Management System (AMS-https://ams.ukbiobank.ac.uk/ams/).

**Funding:** YingShan was supported by National Natural Science Foundation of China (82204148), Shenzhen Science and Technology Innovation Commission Scientific Research Fund Projects (JCYJ20220531094401003 and JCYJ20190809111017163), Shenzhen's Sanming Project (SZSM201612061), and Scientific Research Foundation of Peking University Shenzhen Hospital (KYQD2022203). The funders had no role in the study design; in the collection, analysis, and interpretation of data; in the writing of the report; or in the decision to submit the article for publication.

**Competing interests:** The authors have declared that no competing interests exist.

## Conclusion

Daytime napping was associated with an increased risk of CKD. However, the absence of conclusive evidence did not indicate a connection between daytime napping and ESKD.

## Introduction

Chronic kidney disease (CKD) is a growing public health concern worldwide, affecting approximately 10–15% of the adult population, and its prevalence is expected to increase further in the near future [1, 2]. Therefore, with the rising incidence of CKD, it is beneficial to identify potential risk factors as a preventive measure. Emerging evidence suggests that chronic sleep disorders may contribute to metabolic dysfunction. Inappropriate sleep duration has been linked to adverse health outcomes, such as diabetes [3, 4], obesity [5], hypertension [6, 7], osteoporosis [8], cardiovascular disease [9, 10], stroke [11], and total mortality [12]. Studies have also suggested that sleep disturbances may negatively impact kidney function. However, existing research has primarily focused on sleep duration and quality, and more comprehensive investigations are necessary to better understand the association between sleep and kidney function, particularly with regard to daytime napping and CKD. Recent studies have shown correlations between daytime napping and various health outcomes, including all-cause mortality [10], cardiovascular disease [13], metabolic syndrome [14], diabetes [3, 15, 16] and inflammation [17]. However, the relationship between daytime napping and the risk of CKD or end-stage kidney disease (ESKD) remains unclear.

To our knowledge, only three studies have explored the relationship between daytime napping and kidney disease. Two cross-sectional studies have suggested that increased daytime sleep duration is significantly associated with a lower estimated glomerular filtration rate (eGFR) and a higher urinary albumin creatinine ratio (UACR) in diabetics, as well as a higher risk of renal hyperfiltration (RHF) among individuals who nap compared to those who do not [18, 19]. Another longitudinal (REACTION) study from China found that daytime napping is associated with the incidence of microalbuminuria [20]. Prospective cohort studies are thereafter necessary to confirm the association between napping and CKD. Short or long sleep duration assessed by self-reported or wrist actigraphy has been associated with a higher risk of CKD [21–24]. It is unknown whether napping can compensate for the lack of nighttime sleep provided that the total sleep duration is the same. Furthermore, no study has investigated the association between daytime napping and ESKD.

This study aimed to investigate the association between daytime napping and the incidence of CKD and ESKD. We also sought to examine whether sleeping duration modified this relationship.

## Methods

### Study design and setting

This study was conducted using UK Biobank databases (2006–2021). Approval for the UK Biobank study was granted by the National Information Governance Board for Health and Social Care in England and Wales, the Community Health Index Advisory Group in Scotland, and the North West Multicenter Research Ethics Committee. Written informed consent was obtained from all participants. The Strengthening the Reporting of Observational Studies in Epidemiology (STROBE) statement is provided in (S1 Checklist) [25].

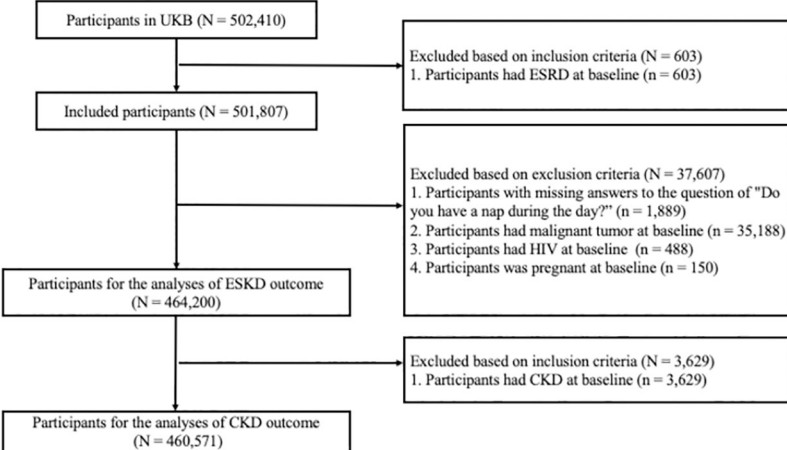

**Fig 1. Participants included and excluded from this study.** CKD, Chronic kidney disease; ESKD, End-stage kidney disease.

## Data sources

Patient characteristics, prescription drug use, covariate information, and outcome data were obtained from 22 assessment centers across England, Scotland, and Wales. The datasets were linked using unique encoded identifiers and analyzed in the UK Biobank database. The primary care data, hospital inpatient data, and death register records were coded using the International Classification of Diseases, 10th Revision (ICD-10) system by trained personnel. The personnel only considered physician-recorded diagnoses in a patient's medical chart when assigning codes and did not review or interpret symptoms or test results. Additional information on the databases, variable definitions, and administrative codes were provided in S1 Table.

## Study population

The UK Biobank is a large-scale biomedical database and research resource and its study protocol has been presented previously [26, 27]. As detailed in the flow chart diagram (Fig 1), inclusion criteria required participants to agree to participate in the UK Biobank program and sign the informed consent at baseline. Exclusion criteria included having ESRD at baseline, missing answers to the question "Do you have a nap during the day?", having a malignant tumor or HIV, and being pregnant at baseline. Additional exclusion criteria were applied for the CKD outcome that we excluded those had CKD at baseline.

## Measures

**Exposure factors: Sleep behaviors.** Details about sleep behaviors, encompassing sleep duration and daytime napping, were gathered through a touchscreen questionnaire administered at recruitment. Participants' sleep duration was evaluated by posing the question: how many hours of sleep do you have in every 24 hours including naps? According to the National Sleep Foundation's sleep time duration recommendations, the panel agreed that, for healthy individuals with normal sleep, the appropriate sleep duration for a middle- to older-aged adults was between 7 to 8 hours [28]. The sleep duration of participants was divided into three groups: ≤6h, 7-8h, and ≥9h. Data on daytime sleeping were collected by the question: do you

have a nap during the day? The responses included never/rarely, sometimes, often, and prefer not to answer. In cases of significant variation in sleep behaviors, participants were requested to furnish average data over the preceding month.

**Covariates.** The selection of all covariates was predetermined, guided by their relevance to both napping and CKD [29–38]. They included age, sex, ethnicity, education, Townsend deprivation index, physical activity (metabolic equivalent [MET] scores), smoking status, alcohol intake, hypnotic drug use, history of cardiovascular disease (CVD) including stroke, hypertension, diabetes mellitus, dyslipidemia or hyperlipidemia, C-reactive protein (CRP), waist circumference (WC), eGFR and UACR. A directed acyclic graph (DAG) explaining the potential pathways from exposure to outcomes and include confounders and potential mediators was shown in S1 Fig. Sociodemographic characteristics, life course exposures, and medical history were obtained through touch-screen questionnaires at recruitment. Additionally, physical measurements, blood and urine samples were also collected at baseline. For each participant, eGFR at baseline was calculated using the Chronic Kidney Disease Epidemiology Collaboration (CKD-EPI) equation (shown in S2 Table) on the basis of cystatin C levels measured by the standard enzymatic method. Hypnotics were classified as benzodiazepines, benzodiazepine-like compounds (zolpidem and zopiclone), and miscellaneous medications (including barbiturates, antihistamines and other pharmacological categories such as neuroleptics) [39].

**Outcome ascertainment.** Prevalent CKD was characterized by an eGFR below 60 mL/min per 1.73 m2 or UACR above 30 mg/g at recruitment, or identified through ICD-10 or Office of Population Censuses and surveys-4 (OPCS-4) codes. The date of ESKD was officially compiled by the UK Biobank, as described on the official website (www.ukbiobank.ac.uk). Incident CKD and ESKD cases were identified by ICD Codes in any primary care data, hospital inpatient data, and death register records. Indications of renal replacement therapy cases were obtained on the basis of ICD-10 codes and OPCS-4 codes using hospital inpatient data. Hospital inpatient records were acquired by linking with the Hospital Episode Statistics for England, Scottish Morbidity Records for Scotland, and Patient Episode Database for Wales. The identification of CKD and ESKD-related deaths was accomplished through linking with the death registry. The follow-up time was calculated from the date of attending the assessment center to the date reported for diagnosis of CKD events, ESKD events, death, loss to follow-up, or end of the follow-up (November 12, 2021), whichever occurred first.

**Statistical analyses.** The characteristics of the participants were first presented by the categories of daytime napping (never or rarely/sometimes/usually). We illustrated the baseline characteristics before and after the imputation. Imputation of the phenotypic data was carried out using the R package Mice [40].

With the participants met the inclusion and the exclusion criteria for CKD and ESKD, respectively, we investigated the association between daytime napping and the risk of CKD and ESKD using multivariable Cox proportional hazards regression models. Participants who reported never or rarely napping served as the reference group. The Schoenfeld residuals were used to test the proportional hazards assumption of the Cox models. In model 1, adjustments were made for the following potential confounders: age at recruitment (continuous), sex (men, women), ethnicity (White, Asian, Black, others), education (level 1, "uneducated"; level 2, "O levels/GCSEs or equivalent" or "CSEs or equivalent"; level 3, "A levels/AS levels or equivalent" or "NVQ or HND or HNC or equivalent" or "Other professional qualifications"; level 4, "College or University degree"), Townsend deprivation index (continuous), physical activity (MET scores, continuous), smoking status (never, previous, and current) and alcohol intake (never, previous, and current). In model 2, additional adjustments included hypnotic drug use (yes/no), history of CVD, hypertension, diabetes mellitus, dyslipidemia, CRP (continuous) and WC (continuous). In model 3, further adjustments incorporated eGFR (continuous) and UACR

(continuous). The P value for trend was calculated treating the napping frequency as ordinal values in the models.

To assess the robustness of our study, several sensitivity analyses were performed. The models were comprehensively adjusted, mirroring the adjustments made in Model 3. First, we performed analyses exclusively on participants without a history of CVD, hypertension, diabetes mellitus, obstructive sleep apnea (OSA) and dyslipidemia at baseline. Second, to mitigate the impact of reverse causality on the observed associations, we reanalyzed the data focusing on participants with more than 2 years of follow-up. Third, to account for the competing risks of death, we employed the Fine-Gray sub distribution method.

To evaluate potential effect modification, we stratified the individuals into 7-8h, ≤6h, and ≥9h of sleep groups by total sleeping duration for the risk of CKD and ESRD, and the models were further adjusted for the model 3. Multiplicative interaction was tested by including interaction terms between daytime napping and sleeping duration. We also assessed additive interaction by calculating the relative excess risk due to interaction (RERI) and the attributable proportion (AP). The 95% CI of RERI and AP were generated by 1,000 bootstrap resampling iterations.

All analyses were conducted utilizing R version 4.0.3. The following packages were applied: 'mice', 'dplyr', 'dplyr', 'stringr', 'survival', 'survminer', 'cmprsk', 'riskRegression', 'car', 'Publish', 'rms', 'tableone', 'mice', 'forestplot', and 'boot'. Statistical significance was defined as a two-sided P-value less than 0.05.

## Results

### Baseline characteristics

In the UK Biobank study, 502,410 participants were initially investigated, and 460,571 individuals were ultimately included in this study (Fig 1). The median baseline age of the participants was 70.9 (interquartile range: 65.0–87.8) years, and 57.9% were female. Distributions of the baseline characteristics according to daytime napping were shown in Table 1, while the checked distributions were presented in S3 Table. Individuals who reported napping were more likely to be men, older, with lower educational attainment, residing in more deprived areas, past or current smokers, individuals who never or previously consumed alcohol, physically inactive, and had higher levels of WC. Additionally, they exhibited a heightened prevalence of hypertension, diabetes, cardiovascular disease, diabetes mellitus, and dyslipidemia. They were more prone to using hypnotic drugs, had lower eGFR, and higher levels of UACR and CRP at baseline. These factors were taken into account and adjusted for in subsequent analyses.

### Association of daytime napping with incident CKD

Over a mean follow-up duration of 11.1 years (SD 2.2), we recorded 28,330 newly diagnosed cases of CKD. In the full model, with further adjustments, although the estimates showed a meaningful attenuation, the association between napping and elevated risk of CKD remained consistent (P for trend = .004). Compared with the reference group, the HRs (95% CIs) for CKD in model 3 were 1.07 (1.04 to 1.11) for the participants who sometimes took naps and 1.09 (1.03 to 1.16) for those who usually took naps (Fig 2).

### Association of daytime napping with ESKD

Over a mean follow-up period of 11.1 years (SD 2.2), we observed 927 cases of ESRD. In the full model, after adjusting for covariates in model 3, relative to the reference group, the HRs

**Table 1. Baseline characteristics of the participants according to daytime napping in the UK Biobank.**

| | | | Daytime napping | | | |
|---|---|---|---|---|---|---|
| | Sample size (% missing) | Never/rarely (n = 262,567) | Sample size (% missing) | Sometimes (n = 177,268) | Sample size (% missing) | Usually (n = 24,365) |
| Age (y) | | 56.0 [48.0, 62.0] | | 59.0 [51.0, 64.0] | | 61.0 [54.0, 65.0] |
| Men (%) | | 108,526 (41.3) | | 88,770 (50.1) | | 15,964 (65.5) |
| Ethnicity | 805 (0.3) | | 706 (0.4) | | 112 (0.5) | |
| Asian | | 5283 (2.0) | | 4716 (2.7) | | 758 (3.1) |
| Black | | 3556 (1.4) | | 3626 (2.1) | | 428 (1.8) |
| White | | 249,191 (95.2) | | 165,329 (93.6) | | 22,648 (93.4) |
| Others | | 3732 (1.4) | | 2891 (1.6) | | 419 (1.7) |
| Educational status | 4339(1.7) | | 3478(2.0) | | 481 (2.0) | |
| Level1 | | 36,179 (14.1) | | 35,122 (20.2) | | 6010 (25.2) |
| Level2 | | 70,986 (27.5) | | 46,219 (26.6) | | 5782 (24.2) |
| Level3 | | 59,310 (23.0) | | 4,0731 (23.4) | | 5435 (22.8) |
| Level4 | | 91,753 (35.5) | | 51,718 (29.8) | | 6657 (27.9) |
| TDI | | -2.27 [-3.71, 0.23] | | -1.96 [-3.55, 0.88] | | -1.62 [-3.37, 1.53] |
| Smoking status | 847 (0.3) | | 735 (0.4) | | 117 (0.5) | |
| Never | | 150,974 (57.7) | | 92,115 (52.2) | | 11,101(45.8) |
| Previous | | 85,653 (32.7) | | 63,533 (36.0) | | 9551 (39.4) |
| Current | | 25,093 (9.6) | | 20,885 (11.8) | | 3596 (14.8) |
| Alcohol consumption | 261 (0.1) | | 234 (0.1) | | 46 (0.2) | |
| Never | | 10,484 (4.0) | | 8493 (4.8) | | 1420 (5.8) |
| Previous | | 7727 (2.9) | | 7204 (4.1) | | 1575 (6.5) |
| Current | | 244,095 (93.1) | | 161,337 (91.1) | | 21,324 (87.7) |
| MET(m/w) | | 1782.0 [826.5, 3546.0] | | 1758.0[782.0, 3586.9] | | 1666.0[655.0, 3624.0] |
| WC(cm) | | 88.0 [79.0, 97.0] | | 92.0[83.0, 101.0] | | 96.00 [87.00, 105.00] |
| Hypertension | 2 (0.0) | 58,505 (22.3) | 0 (0.0) | 54,205 (30.6) | | 9357 (38.4) |
| Diabetes Mellitus | 3 (0.0) | 8922 (3.4) | 3 (0.0) | 12,144 (6.9) | | 2679 (11.0) |
| CVD | | 22,380 (8.5) | | 23,104 (13.0) | | 4872 (20.0) |
| Dyslipidemia | | 116,102 (44.2) | | 85,927 (48.5) | | 12,449 (51.1) |
| eGFR | | 91.93 [80.09, 102.62] | | 87.14 [75.40, 99.20] | | 83.68[71.89, 96.22] |
| UACR (mg/g) | 18,011 (6.9) | | 11,123 (6.3) | | 1506 (6.2) | |
| <30 | | 235,111 (96.1) | | 157,364 (94.7) | | 21,255 (93.0) |
| 30–300 | | 8775 (3.6) | | 7936 (4.8) | | 1435 (6.3) |
| ≥300 | | 670 (0.3) | | 845 (0.5) | | 169 (0.7) |
| CRP | | 1.20 [0.60, 2.49] | | 1.46 [0.72, 3.02] | | 1.62 [0.78, 3.39] |
| Hypnotic drug use | | 5876 (2.2) | | 4870 (2.7) | | 1029 (4.2) |
| CKD event | | 12,582 (4.8) | | 13,193 (7.4) | | 2555 (10.5) |
| ESKD event | | 341 (0.1) | | 470 (0.3) | | 116 (0.5) |

TDI, Townsend deprivation index; MET, metabolic equivalent task; WC, waist circumference; CVD, cardiovascular disease; eGFR, estimated glomerular filtration rate; UACR, urinary protein creatinine ratio; CRP, C-Reactive Protein; CKD, chronic kidney disease; ESKD, end-stage kidney disease. level 1, "uneducated"; level 2, "O levels/GCSEs or equivalent" or "CSEs or equivalent"; level 3, "A levels/AS levels or equivalent" or "NVQ or HND or HNC or equivalent" or "Other professional qualifications"; level 4, "College or University degree".

Data are presented as mean ± standard deviation or median (interquartile range) for continuous variables and number (percentage) for categorical variable.

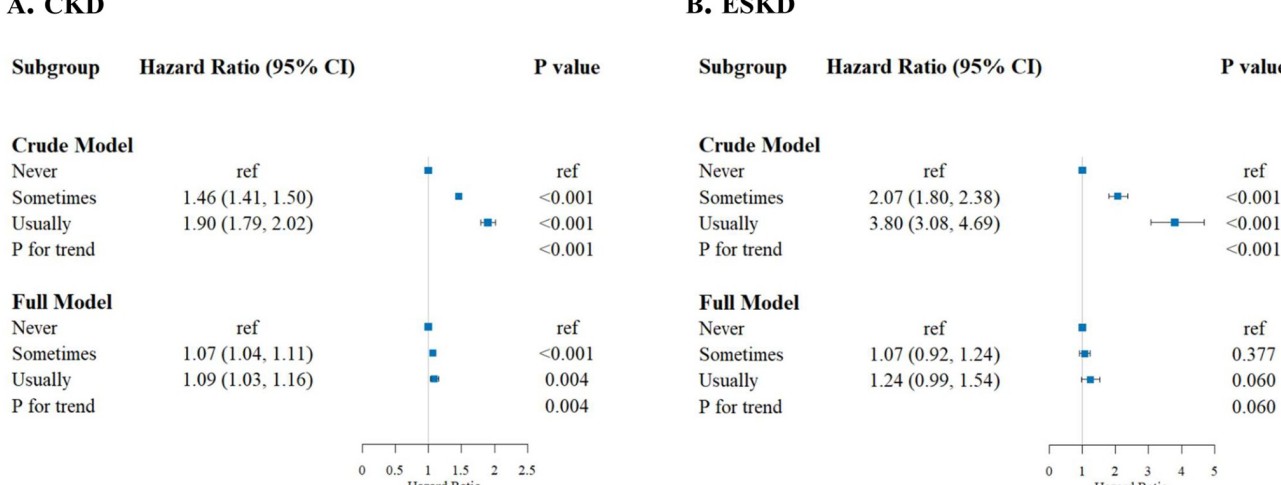

**Fig 2. Risk of incident CKD and ESKD according to daytime napping in the UK Biobank study.** TDI, Townsend deprivation index; MET, metabolic equivalent; WC, waist circumference; CVD, cardiovascular disease; CRP, C-reactive protein; eGFR, estimated glomerular filtration rate; UACR, urinary albumin creatinine ratio; [a]Crude Model: Unadjusted variates; [b]Full Model: Hazard ratios (HRs) were adjusted for age, sex, ethnicity, educational status, TDI, smoking status, alcohol consumption, MET scores, WC, hypnotic drug use, history of CVD, hypertension, diabetes mellitus and dyslipidemia, CRP, basic eGFR and UACR.

(95% CIs) for ESKD were 1.07 (0.92 to 1.24) in the group that occasionally naps and 1.24 (0.99 to 1.54) in the group that usually naps (as shown in Fig 2). A correlation was observed between napping and an increased risk of ESKD as well. Nevertheless, the P-values (0.377 and 0.060, respectively) did not allow to exclude the effect of confounding factors embedded in the data. Therefore, an association between napping and elevated risk of ESKD could not be asserted.

## Interactions between daytime napping and sleep duration and sensitivity analyses

We did not observe any indication of an additive interaction between daytime napping and sleep duration, as evidenced by 95% CI of the RERI and AP covering the null. Specifically, participants who neither engaged in napping nor slept for 7–8 hours exhibited the lowest risk of CKD and were consequently designated as the reference group. In comparison to the reference group, individuals who sometimes or usually napped and slept for $\geq$9 hours had RERI values of 0.006 (95% CI, -0.094 to 0.07) and -0.265 (95% CI, -0.094 to 0.07), respectively. The corresponding AP values were 0.005 (95% CI, -0.111 to 0.116) and -0.257 (95% CI, -0.475 to -0.068), respectively. Comparable findings were observed for individuals with $\leq$6 hours of sleep (Tables 2 and 3). We did not detect any multiplicative interaction between daytime napping and sleep duration (as indicated in Table 3), either. For the ESKD outcome, no evidence of either multiplicative or additive interaction was observed, as presented in Tables 4 and 5. In the sensitivity analyses, the results were generally consistent with the main analyses, as detailed in S4–S6 Tables.

## Discussion

In this extensive prospective study involving a middle-to-older-aged British population with an 11 years of follow-up, daytime napping was identified as a factor associated with an elevated risk of incident CKD. However, insufficient evidence indicated that daytime napping was

**Table 2. Modification of the effect of daytime napping on CKD by sleep duration.**

| | | Sleep duration | | | | | | | |
|---|---|---|---|---|---|---|---|---|---|
| | | 7–8 (h) | | < = 6 (h) | | | > = 9 (h) | | |
| | | N with/ without CKD event | RR (95% CI) | N with/ without CKD event | RR (95% CI) | RRS (95% CI) | N with/ without CKD event | RR (95% CI) | RRs (95% CI) |
| Daytime napping | Never/ Rarely | 5265/16,5879 | 1 | 2401/59,776 | 1.115 (1.062, 1.171) P<0.001 | 1.115 (1.062, 1.171) P<0.001 | 511/10,723 | 1.150 (1.050, 1.259) P = 0.003 | 1.150 (1.050, 1.259) P = 0.003 |
| | Sometimes | 4700/101,675 | 1.077 (1.034, 1.120) P<0.001 | 2009/36,092 | 1.179 (1.119, 1.242) P<0.001 | 1.095 (1.039, 1.154) P = 0.001 | 883/13,242 | 1.234 (1.148, 1.327) P<0.001 | 1.147 (1.067, 1.232) P<0.001 |
| | Usually | 701/11,053 | 1.165 (1.075, 1.261) P<0.001 | 275/3959 | 1.201 (1.063, 1.357) P = 0.003 | 1.031 (0.897, 1.186) P = 0.668 | 293/4433 | 1.040 (0.924, 1.171) P = 0.517 | 0.893 (0.779, 1.024) P = 0.104 |

RRs are adjusted for baseline age, sex, race, education, TDI, smoking status, alcohol consumption, MET scores, hypertension, diabetes mellitus, CVD, dyslipidemia, hypnotic drug use, WC, CRP, eGFR and UACR.

associated with ESKD. Furthermore, this study demonstrated that the duration of sleep did not modify the effect of daytime napping on the risks of CKD and ESKD.

In 2017, nearly 700 million individuals were affected by all-stage CKD, surpassing the prevalence of conditions such as diabetes, osteoarthritis, chronic obstructive pulmonary disease, asthma, or depressive disorders [41]. As per the Global Burden of Disease study, CKD is positioned as the 12th leading cause of death among 133 conditions [42]. Kidney disease has a profound impact on global health, acting as a direct contributor to global morbidity and mortality and a notable risk factor for CVD. Despite its preventable and treatable nature, CKD demands heightened consideration in global health policy decision-making [2].

Previous studies have focused on the association of sleeping duration or quality and CKD. A study conducted in Japan and a recent analysis from the Nurses' Health Study both indicate that short self-reported sleep duration is linked to a rapid decline in renal function [21, 43]. Studies carried out in Taiwan and the US, involving adults with predialysis CKD, have highlighted a correlation between CKD and both short sleep duration and poor sleep quality [44, 45]. An 11-year prospective study in France conducted among the elderly has showed that excessive daytime sleepiness, restless legs syndrome and apnea hypopnea index are risk factors for renal function alteration at a very early stage of decline. Few studies have examined the association between daytime napping and CKD. Until recently, two cross-sectional investigations suggest that increased daytime sleep duration is significantly associated with reduced eGFR, higher UACR, and a higher risk of RHF among those who took naps longer than 1.5 hours per day compared with those who did not nap [18, 19]. Another longitudinal study from China has found that daytime napping is positively associated with the incidence of microalbuminuria, but is not correlated between daytime napping and reduced eGFR [20]. In contrast, a study from iranian ravansar cohort demonstrated that after adjusting for confounding factors, no statistically significant association of daytime napping with CKD was observed among Iranian adults in the Kurdish region [46]. Even after accounting for potential confounding factors, our study still indicated an association between daytime napping and an increased risk of developing CKD. Despite a weakened association, the p-value for the trend (= .040) remains statistically significant. The variations in these findings might stem from differences among participants in terms of race, lifestyle, and health. Consequently, future studies should take into account the varying risk levels for CKD across diverse racial and ethnic groups.

**Table 3. Joint effects of daytime napping and sleep duration on the risk of incident CKD.**

| | | Sleep duration | | | | | |
|---|---|---|---|---|---|---|---|
| | | < = 6 h | | | > = 9 h | | |
| | | RERI (95% CI) | AP (95% CI) | Multiplicative scale interaction (95% CI) | RERI (95% CI) | AP (95% CI) | Multiplicative scale interaction (95% CI) |
| Daytime napping | Sometimes | -0.013 (-0.094, 0.07) | -0.011(-0.083, 0.055) | 0.982 (0.9144, 1.0543) | 0.006 (-0.094, 0.07) | 0.005 (-0.111, 0.116) | 0.996 (0.8873, 1.1187) |
| | Usually | -0.073 (-0.242, 0.11) | -0.06 (-0.234, 0.08) | 0.929 (0.8012, 1.0767) | -0.265 (-0.094, 0.07) | -0.257(-0.475, -0.068) | 0.782 (0.6638, 0.9217) |

RERI, relative excess risk of interaction; AP, attributable proportion.

Measure of effect modification on additive scale: RERI (95% CI) and AP(95%CI); Measure of effect modification on multiplicative scale: ratio of RRs(95% CI).

The specific pathophysiological mechanisms underlying the connection between daytime napping and CKD remain poorly understood. The mechanisms that potentially link napping with CKD need further clarification. Napping may predispose to incident CKD by increasing the risk of developing established CKD risk factors. For example, a systematic review suggested that among older adults aged >60 years, dose-response associations of daytime napping with higher odds of diabetes, dyslipidemia, metabolic syndrome and mortality were observed, starting from 0 min/d [47]. In a study during a median follow-up of 8.1 years, regular long (> 60 min) midday nap was associated with an increased hazard ratio of cardiac events [13]. In contrast, an observational prospective cohort study from Swiss population found that subjects who nap once or twice per week have a lower risk of incident CVD events, while no association was found for more frequent napping or napping duration [48].

These studies additionally indicated that napping could be a modifiable factor influencing these well-established CKD risk factors. However, in our cohort, adjusting for the development of hypertension, diabetes, CVD and dyslipidemia at baseline did not materially change effect estimates for the risk of incident CKD with daytime napping. This suggests that daytime napping is associated with the risk of incident CKD via a mechanism independent of these factors. Short sleep and poor sleep quality can lead to daytime naps. Conversely, daytime naps can delay sleep onset, decreasing the homeostatic sleep drive [49, 50]. A Mendelian randomization study about short or long sleep duration and CKD showed causal effects of short sleep

**Table 4. Modification of the effect of daytime napping on ESKD by sleep duration.**

| | | Sleep duration | | | | | | | |
|---|---|---|---|---|---|---|---|---|---|
| | | 7–8 (h) | | < = 6 (h) | | | > = 9 (h) | | |
| | | N with/ without ESKD event | RR (95% CI) | N with/ without ESKD event | RR (95% CI) | RRS (95% CI) | N with/ without ESKD event | RR (95% CI) | RRs (95% CI) |
| Daytime napping | Never/ Rarely | 214/182,590 | 1 | 107/67,190 | 1.114 (0.881, 1.409) P = 0.368 | 1.114 (0.881, 1.409) P = 0.368 | 20/12,446 | 1.027 (0.648, 1.628) P = 0.910 | 1.027 (0.648, 1.628) P = 0.910 |
| | Sometimes | 282/117,454 | 1.076 (0.896, 1.291) P = 0.433 | 124/42,921 | 1.123 (0.895, 1.411) P = 0.316 | 1.044 (0.844, 1.293) P = 0.690 | 64/16,423 | 1.245 (0.934, 1.659) P = 0.136 | 1.157 (0.880, 1.522) P = 0.297 |
| | Usually | 47/13,488 | 1.109 (0.803, 1.530) P = 0.531 | 23/4989 | 1.118 (0.720, 1.736) P = 0.620 | 1.008 (0.611, 1.665) P = 0.974 | 46/5772 | 1.694 (1.218, 2.357) P = 0.002 | 1.528 (1.015, 2.301) P = 0.042 |

RRs are adjusted for baseline age, sex, race, education, TDI, smoking status, alcohol consumption, MET scores, hypertension, diabetes mellitus, CVD, dyslipidemia, hypnotic drug use, WC, CRP, eGFR and UACR.

**Table 5. Joint effects of daytime napping and sleep duration on the risk of incident ESKD.**

| | | Sleep duration | | | | | |
| | | < = 6 h | | | > = 9 h | | |
| | | RERI (95% CI) | AP (95% CI) | Multiplicative scale interaction | RERI (95% CI) | AP (95% CI) | Multiplicative scale interaction |
|---|---|---|---|---|---|---|---|
| Daytime napping | Sometimes | -0.083 (-0.481, 0.263) | -0.073 (-0.44, 0.192) | 0.923 (0.6732, 1.2650) | 0.172(-0.481, 0.263) | 0.134 (-0.353, 0.508) | 1.152 (0.6755, 1.9659) |
| | Usually | -0.124 (-0.803, 0.512) | -0.117(-1.012, 0.34) | 0.889 (0.5102, 1.5489) | 0.495 (-0.481, 0.263) | 0.337 (-0.157, 0.707) | 1.512 (0.8137, 2.8093) |

RERI, relative excess risk of interaction; AP, attributable proportion.

Measure of effect modification on additive scale: RERI (95% CI) and AP (95% CI); Measure of effect modification on multiplicative scale: ratio of RRs (95% CI).

duration on CKD had been demonstrated [51]. Two studies from the United States have suggested that short sleep duration and less efficient sleep can lead to napping and napping can lead to short and less efficient sleep [50, 52]. These findings may be the potential mechanisms underlying an association of napping with incidence of CKD.

The researches have found associations between OSA, poor sleep quality, and both napping and CKD [53–55]. OSA may serve not only as a mediating factor but also as a potential confounder. This is due to the fact that OSA may lead to poor sleep quality and daytime sleepiness, which in turn could increase the duration of daytime napping. To address this issue more thoroughly, we have revised our Sensitivity Analysis 1, in which we additionally excluded participants diagnosed with OSA at the baseline. The results from this sensitivity analysis aligned with those of our main analysis. Despite an increase in the P-value resulting from a reduction in sample size, the effect size and direction remained consistent. This suggested the robustness of our findings despite the exclusion of OSA cases.

To our knowledge, this was the first prospective study to explore the relationship between daytime napping and ESKD. After adjusting for confounding factors, the P-value for trend (= 0.060) was no longer statistically significant. Therefore, it could not be conclusively stated that a relationship exists between napping and an increased risk of ESKD. Daytime napping is a prevalent social practice adopted by many individuals as part of their routine. Taking a nap can relieve fatigue and improve afternoon work efficiency. However, inappropriate napping can result in feelings of grogginess and emotional fatigue. Furthermore, a growing body of research suggests that prolonged napping elevates the risk of chronic diseases among middle-aged and elderly individuals. Certainly, the existence of a napping habit in a population unaccustomed to such behavior could potentially signal important health implications. It is noteworthy that participants in our study who engaged in napping tended to be older and exhibited a less favorable health profile. Consequently, the likelihood of residual confounding due to declining health cannot be dismissed. There is substantial evidence indicating that napping may promote the development and exacerbate the severity of three important risk factors for CKD, namely type 2 diabetes, hypertension and obesity. Numerous studies have also established associations between Body Mass Index [56], hypertension [57], and type 2 diabetes [58] with ESKD. More precisely, these factors might serve as mediators in the relationship between napping and the risk of ESKD. After adjusting for these factors, no statistically significant association of daytime napping with ESKD was observed, indicating that these factors might confound the observed associations between daytime napping and ESKD.

Our study boasted several strengths, including its prospective cohort design, large sample size, extensive data collection, and assessing the clinically relevant hard outcomes such as CKD

or ESKD, which allowed us to meticulously adjust for a wide range of potential confounders. On the other hand, our study has several limitations. First, as an observational cohort study, we cannot conclude causal inference between daytime napping and CKD or ESKD. Second, our sample consisted of British White middle-aged or older adults, and the definition and cultural acceptance of napping may differ in other ethnic or racial groups, limiting the generalizability of our results. Third, we used a self-reported questionnaire to collect information about napping frequency, rather than objective measurements. However, previous studies have demonstrated good correlation between self-reported and objective measurements [59]. In future studies, measuring napping duration may provide additional insights into the association between napping and CKD or ESKD. Fourth, despite extensive adjustment for the potential confounders, residual confounding cannot be completely ruled out in observational studies. Last, the diagnosis of CKD and ESKD mainly depends on the ICD codes, which may result in missed diagnoses, especially for CKD. Future studies with repeated measurements of renal function indicators are needed to better assess the association between changes in napping behavior and risk of CKD or ESKD.

## Conclusions

In conclusion, self-reported daytime napping was positively associated with an increased incident CKD risk, suggesting that napping may be a potential risk factor for the onset of CKD. However, the specific mechanism by which daytime napping induces the occurrence of CKD is not well understood; Future work is needed to explore the specific mechanisms and assess whether the observed relationship between napping with incident CKD is causal.

## Supporting information

**S1 Checklist. STROBE statement.**
(PDF)

**S1 Fig. Directed acyclic graph illustrating the relationships between exposures, outcomes, potential mediators and potential confounders in the study.** eGFR, estimated glomerular filtration rate; UACR, urinary protein creatinine ratio; CKD, chronic kidney disease; ESKD, end-stage kidney disease.
(TIF)

**S1 Table. ICD encoding of all variables in the UK Biobank.**
(PDF)

**S2 Table. The Chronic Kidney Disease Epidemiology Collaboration (CKD-EPI).** Equation on the basis of cystatin C levels for estimating GFR. GFR, glomerular filtration rate. The CKD-EPI cystatin C equation (2012) can be expressed as a single equation: $133 \times \min(Scys/0.8, 1) - 0.499 \times \max(Scys/0.8, 1) - 1.328 \times 0.996$ Age $[\times 0.932$ if female], where Scys is serum cystatin C, min indicates the minimum of $Scr/\kappa$ or 1, and max indicates the maximum of $Scys/\kappa$ or 1.
(PDF)

**S3 Table. Baseline characteristics of the participants according to daytime napping with data after check in the UK Biobank.** TDI, Townsend deprivation index; MET, metabolic equivalent task; WC, waist circumference; CVD, cardiovascular disease; eGFR, estimated glomerular filtration rate; UACR, urinary protein creatinine ratio; CRP, C-Reactive Protein; CKD, chronic kidney disease; ESKD, end-stage kidney disease. level 1 = "uneducated", level 2 = "O levels/GCSEs or equivalent" or "CSEs or equivalent", level 3 = "A levels/AS levels or equivalent" or "NVQ or HND or HNC or equivalent" or "Other professional qualifications",

level 4 = "College or University degree". Data are presented as mean + standard deviation or median (interquartile range) for continuous variables and number (percentage) for categorical variables.
(PDF)

**S4 Table. Association between daytime napping and incident CKD or ESKD in participants who without hypertension, diabetes mellitus, CVD, and OSA dyslipidemia at baseline.** CKD, chronic kidney disease; ESKD, end-stage kidney disease; CVD, cardiovascular disease; OSA, obstructive sleep apnea. Statistical analysis using Cox regression. Results are expressed as multivariable-adjusted hazard ratios and (95% confidence interval). All analyses are adjusted for basic age, sex, ethnicity, educational status, TDI, smoking status, alcohol consumption, MET scores, WC, hypnotic drug use, history of CVD, hypertension, diabetes mellitus and dyslipidemia, CRP, basic eGFR and UACR.
(PDF)

**S5 Table. Association between daytime napping and incident CKD or ESKD in participants whose outcome events occurred after the first 2 years of follow-up.** Statistical analysis using Cox regression. Results are expressed as multivariable-adjusted hazard ratios and (95% confidence interval). All analyses are adjusted for basic age, sex, ethnicity, educational status, TDI, smoking status, alcohol consumption, MET scores, WC, hypnotic drug use, history of CVD, hypertension, diabetes mellitus and dyslipidemia, CRP, basic eGFR and UACR.
(PDF)

**S6 Table. Association between daytime napping and incident CKD or ESKD in competing-risk models.** Statistical analysis using Cox regression. Results are expressed as multivariable-adjusted hazard ratios and (95% confidence interval). All analyses are adjusted for basic age, sex, ethnicity, educational status, TDI, smoking status, alcohol consumption, MET scores, WC, hypnotic drug use, history of CVD, hypertension, diabetes mellitus and dyslipidemia, CRP, basic eGFR and UACR.
(PDF)

## Acknowledgments

We give our special thanks to the staff of nephrology in the First Affiliated Hospital of Soochow University for their cooperation and support. We also thank the members of clinical research academy in the Peking University Shenzhen Hospital and the UK Biobank (https://www.ukbiobank.ac.uk/) for publicly sharing the data we used in our analysis.

## Author Contributions

**Conceptualization:** Qinjun Li, Ling Wang, Liang Dai, Sen Kan, Xiaoyan Huang, Guoyuan Lu.

**Data curation:** Ying Shan.

**Formal analysis:** Ying Shan, Jingchi Liao, Yanling Wei.

**Funding acquisition:** Ying Shan.

**Investigation:** Qinjun Li, Ling Wang, Sen Kan, Xiaoyan Huang, Guoyuan Lu.

**Methodology:** Ying Shan, Jingchi Liao, Liang Dai, Xiaoyan Huang.

**Project administration:** Qinjun Li, Xiaoyan Huang, Guoyuan Lu.

**Resources:** Xiaoyan Huang.

**Supervision:** Jianqing Shi, Xiaoyan Huang.

**Validation:** Qinjun Li, Ling Wang, Sen Kan, Xiaoyan Huang, Guoyuan Lu.

**Visualization:** Qinjun Li, Ying Shan, Jingchi Liao, Xiaoyan Huang, Guoyuan Lu.

**Writing – original draft:** Qinjun Li.

**Writing – review & editing:** Qinjun Li, Ying Shan, Ling Wang, Xiaoyan Huang, Guoyuan Lu.

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
