## [Decision Letter · Decision Letter 0]

4 Oct 2023

PONE-D-23-22161Association of daytime napping with incidence of chronic kidney disease and end-stage kidney disease: A prospective observational studyPLOS ONE

Dear Dr. Lu,

Thank you for submitting your manuscript to PLOS ONE. After careful consideration, we feel that it has merit but does not fully meet PLOS ONE’s publication criteria as it currently stands. Therefore, we invite you to submit a revised version of the manuscript that addresses the points raised during the review process.

We look forward to receiving your revised manuscript.

Kind regards,

Giuseppe Remuzzi

Academic Editor

PLOS ONE

Journal Requirements:

- https://doi.org/10.1016/j.mayocp.2021.08.028

- 10.1016/j.numecd.2016.06.006

- http://dx.doi.org/10.3109/07420528.2012.741171

- https://doi.org/10.1053%2Fj.ajkd.2012.04.027

- https://backend.orbit.dtu.dk/ws/files/220920456/1_s2.0_S0140673620300453_main.pdf

In your revision ensure you cite all your sources (including your own works), and quote or rephrase any duplicated text outside the methods section. Further consideration is dependent on these concerns being addressed.

**Additional Editor Comments:**

**The manuscript focuses on a topic of potential interest. However, the study has major shortcomings that preclude sound conclusions. To mention some of them: i) lack of clear explanation why the Hazard Ratio (HR) value decreases from 1.46 to 1.07 for individuals who sometimes took a nap and HR value decreases from 1.90 to 1.09 for those who usually took a nap when the cruel model (subgroup A-CKD) is adjusted for model 3 (see figure 2); ii) lack of major details regarding the application of the R software; iii) lack of the dataset applied for building multivariable COX proportional hazard regression models; iv) lack of discussion on the potential role of obstructive sleep apnea (OSA) as the potential mediator that explains the association between daytime sleepiness and CKD or ESKD.**

Reviewers' comments:

Reviewer's Responses to Questions

**Comments to the Author**

1. Is the manuscript technically sound, and do the data support the conclusions?

Reviewer #1: Yes

Reviewer #2: Partly

2. Has the statistical analysis been performed appropriately and rigorously? 

Reviewer #1: Yes

Reviewer #2: No

3. Have the authors made all data underlying the findings in their manuscript fully available?

Reviewer #1: Yes

Reviewer #2: Yes

4. Is the manuscript presented in an intelligible fashion and written in standard English?

Reviewer #1: Yes

Reviewer #2: Yes

5. Review Comments to the Author

Reviewer #1: This is a generally well-written manuscript based on data collected prospectively in the UK Biobank Study. The association of daytime napping with CKD and ESKD is positive based on robust cox regression analysis despite adjustment for potential confounders. The authors also examined interaction of daytime sleepiness and sleep duration and did not find significant interaction between these two variables.

One of the major criticisms this reviewer has is that the authors do not adequately speculate the potential mechanisms in the discussion section. They do not explicitly call out the potential role of obstructive sleep apnea (OSA) as the potential mediator that explains the association between daytime sleepiness and CKD or ESKD. Daytime sleepiness and therefore napping during the day are often symptoms of underlying OSA with many negative consequences.

A limitation is the detection of CKD by ICD codes as opposed to lab values. The authors do acknowledge this weakness.

While not absolutely necessary, the manuscript could benefit from inclusion of a DAG (causal diagram) explaining the potential pathways from exposure to outcomes and include confounders and potential mediators in such a diagram.

Reviewer #2: The study shortly entitled “Daytime napping and CKD” corresponding to the Manuscript number “PONE-D-23-22161” investigates: a) the association of daytime napping with the incidence of chronic kidney disease (CKD) and end-stage kidney disease (ESKD); and b) whether sleep duration modifies the association of daytime napping with CKD or ESKD. Through the application of multivariable COX proportional hazard regression models, the just mentioned associations were assessed in 460.571 individuals eventually included in the study.

The authors refer in the result session a consistent association of napping with a higher risk of CKD in a dose-dependent fashion; furthermore they report a consistent association with a higher risk of ESKD in a dose-dependent fashion. On the other side, no evidence of an addictive interaction between daytime napping and sleep duration was found. The authors conclude that “napping was an unrecognized and clinically significant risk factor for incident CKD and ESKD” and that this finding may have important implications for the prevention and management of CKD.

MAJOR CONCERNS:

In the result session, when the cruel model (subgroup A- CKD) is adjusted for model 3 (see figure 2) the Hazard Ratio (HR) value decreases from 1.46 to 1.07 for individuals who sometimes took a nap and HR value decreases from 1.90 to 1.09 for those who usually took a nap. In both cases HR decreases in a meaningful way. The Full model’s HR values indicate a 7% (1.07) and a 9% (1.09) higher risk of CKD in individuals taking sometimes or usually a nap, respectively, when compared with the reference group (no nap). Such values are indicative of a positive association but not a consistent association as defined by the authors across the whole manuscript. (please note that a value of HR of 1.00 indicates no differences among reference group and investigated group). A meaningful decrease of the HR value is shown also for subgroup B (Figure 2) where HR diminishes from 2.07 to 1.07 for individuals who took sometimes a nap; and from 3.80 to 1.24 for those who usually took a nap. In this case, the decrease of the HR values matches an increase in P values exceeding the set limit of 0.05. Therefore the association among daily nap and ESKD looks quite weak to be claimed. Furthermore, the delta among HR values (“sometimes nap” against “usually nap”) of subgroup A and subgroup B is too small to claim “a dose-dependent fashion” for the association.

The meaningful decrease of HR values in both subgroups (A and B) reported after model adjustment may suggest that major causes of CKD and ESKD are embedded in confounders. The weak positive association between day napping and CKD and ESKD is indeed in agreement with the second funding of the authors, that is, a missing addictive interaction between daytime napping and sleep duration.

Finally, day napping cannot be claimed as “an independent risk factor” cause it is not known whether day napping is induced by further confounders that have not been taken into account in the present study. The same authors are aware of that indicating that “the possibility of residual confounding by declining health cannot be ruled out”. It is therefore not known whether managing the day napping would mean a better health for the individuals.

MINOR CONCERNS:

In the “Statisitcal analyses” the authors need to explain how they calculated the “total-person time” in order to estimate the maximum confounders to be taken into account.

Major details regarding the application of the R software should be provided in the same session.

Finally, the reviewer also suggests to provide, as Supplementary data, the dataset applied for building multivariable COX proportional hazard regression models.

6. PLOS authors have the option to publish the peer review history of their article (what does this mean?). If published, this will include your full peer review and any attached files.

Reviewer #1: No

Reviewer #2: No

---

## [Author Response · Author response to Decision Letter 0]

24 Nov 2023

Rebuttal Letter

Nov 12, 2023 

Dear Editor,

Thank you for providing us the invaluable opportunity to revise our manuscript entitled “Association of daytime napping with incidence of chronic kidney disease and end-stage kidney disease: A prospective observational study”. We appreciate the time and guidance regarding the revision by you and reviewers. We have incorporated the reviewer's suggestions to supplement the analysis, making the article more rigorous and enriched in content. We have also made a concerted effort to present our findings more clearly and to provide more precise definitions of relevant concepts to enhance the readability and comprehensibility of the article. For more details, please kindly refer to our point-by-point response presented below. 

All relevant revisions are marked in the resubmitted files. We wish the current version of our manuscript, which has been carefully improved and edited for ensuring the optimal administration and presentation, can meet the high standard of, and be considered for publication in PLoS ONE. Please feel free to contact us if there’s further work should we do. 

Best regards, 

Guoyuan Lu 

Journal Requirements:

R: Thank you for the feedback. We have addressed the style requirements, including file naming, as per PLOS ONE's guidelines.

2. We noticed you have some minor occurrence of overlapping text with the following previous publication(s), which needs to be addressed.

R: Thank you for bringing this to our attention. We have made the necessary revisions to address the occurrence of overlapping text with the previous publication(s).

R: Thank you for pointing that out. We have verified the funding information, and the correct grant numbers for the awards received for our study have been provided in the 'Funding Information' section upon resubmission. (P28)

R: Thank you for bringing this to our attention. We do intend to modify the Data Availability statement. As the data from the UK Biobank requires authorization for access, we are unable to publicly share the data. We have made the necessary changes to the Data Availability statement to reflect this:

“The UK Biobank resource is available to bona fide researchers for health-related research in the public interest. All researchers who wish to access the research resource must register with UK Biobank by completing the registration form in the Access Management System (AMS – https://bbams.ndph.ox.ac.uk/ams/).” (P28)

Additional Editor Comments:

The manuscript focuses on a topic of potential interest. However, the study has major shortcomings that preclude sound conclusions. To mention some of them:

i) lack of clear explanation why the Hazard Ratio (HR) value decreases from 1.46 to 1.07 for individuals who sometimes took a nap and HR value decreases from 1.90 to 1.09 for those who usually took a nap when the cruel model (subgroup A-CKD) is adjusted for model 3 (see figure 2); 

R: Thank you for the great question. The differences between the results of model 3 compared to the crude model are due to the consideration of confounding factors. Confounding factors can distort the relationship between exposure and outcome. A common method to control for confounding is to adjust for the confounding factors in the regression analysis. The STROBE statement recommends reporting results both before and after adjusting for confounding in observational studies. Therefore, we have reported the results before and after controlling for confounding factors as per the STROBE statement, which means presenting the results with and without controlling for confounding factors. Confounding factors can distort the relationship between exposure and outcome, which is why the HR appears larger when not controlling for confounding (crude model), and smaller after controlling for confounding (full model). 

We hope our response addresses the concerns to your satisfaction. Please do not hesitate to let us know if the editors or the reviewers have any further questions or require additional clarification.

ii) Lack of major details regarding the application of the R software;

R: Thank you for pointing the issue out. We have addressed the issues related to the application of R software in the manuscript. The revised paragraph is as follows:

“All analyses were performed using R version 4.0.3. The following packages were applied: 'mice', 'dplyr', 'dplyr', 'stringr', 'survival', 'survminer', 'cmprsk', 'riskRegression', 'car', 'Publish', 'rms', 'tableone', 'mice', 'forestplot', and 'boot'.” (P12)

iii) lack of the dataset applied for building multivariable COX proportional hazard regression models; 

R: Thank you for your thorough review of our manuscript. We have made the following revisions in the 'Statistical analyses' in Method section to address the lack of information regarding the dataset used for building the multivariable Cox proportional hazard regression models. However, according to the regulations of the UK Biobank, we are unable to provide the dataset. The relevant statements are as follows:

“With the participants met the inclusion and the exclusion criteria for CKD and ESKD, respectively, we investigated the association between daytime napping and the risk of CKD and ESKD using multivariable Cox proportional hazards regression models. Participants who reported never or rarely napping served as the reference group.” (P10)

In the 'Study population' section, we stated the inclusion and the exclusion criteria as follows: 

“As detailed in the flow chart diagram (Fig 1), inclusion criteria required participants to agree to participate in the UK Biobank program and sign the informed consent at baseline. Exclusion criteria included having ESRD at baseline, missing answers to the question "Do you have a nap during the day?", having a malignant tumor or HIV, and being pregnant at baseline. Additional exclusion criteria were applied for the CKD outcome that we excluded those had CKD at baseline.” (P7)

In the 'Data availability statement' section, we stated the the regulations of the UK Biobank as follows: 

“The UK Biobank resource is available to bona fide researchers for health-related research in the public interest. All researchers who wish to access the research resource must register with UK Biobank by completing the registration form in the Access Management System (AMS – https://bbams.ndph.ox.ac.uk/ams/).” (P29)

iv) lack of discussion on the potential role of obstructive sleep apnea (OSA) as the potential mediator that explains the association between daytime sleepiness and CKD or ESKD.

R: Thank you for drawing our attention to obstructive sleep apnea (OSA). We acknowledge that our initial analysis and discussion gave limited consideration to OSA. In the context of the relationship between daytime sleepiness and CKD/ESKD, we concur with your observation that OSA serves not only as a mediating factor but also as a potential confounder. This is due to the fact that OSA may lead to poor sleep quality and daytime sleepiness, which in turn could increase the duration of daytime napping. To address this issue more thoroughly, we have revised our Sensitivity Analysis 1, in which we additionally excluded participants diagnosed with OSA at the baseline. The results from this sensitivity analysis aligned with those of our main analysis. Despite an increase in the P-value resulting from a reduction in sample size, the effect size and direction remained consistent. This suggested the robustness of our findings despite the exclusion of OSA cases.

The results of the sensitivity analyses are shown in S4 Table and Rebuttal Table 1. 

Rebuttal Table 1. Association between daytime napping and incident CKD or ESKD in participants who without hypertension, diabetes mellitus, CVD, OSA and dyslipidemia at baseline. 

 CKD

HR (95%CI) 

P value ESKD

HR (95%CI) 

P value

Never ref ref ref ref 

Sometimes 1.06 (1.00-1.14) 0.070 1.41 (0.95-2.08) 0.088

Usually

P for trend 1.06 (0.91-1.23)

1.04 (0.94-1.16) 0.456

0.456 0.99 (0.39-2.52)

0.99 (0.51-1.92) 0.984

0.984

Abbreviations: CKD, chronic kidney disease; ESKD, end-stage kidney disease; CVD, cardiovascular disease; OSA, obstructive sleep apnea. 

Statistical analysis using Cox regression. Results are expressed as multivariable-adjusted hazard ratios and (95% confidence interval). 

All analyses are adjusted for basic age, sex, ethnicity, educational status, TDI, smoking status, alcohol consumption, MET scores, WC, hypnotic drug use, history of CVD, hypertension, diabetes mellitus and dyslipidemia, CRP, basic eGFR and UACR.

Reviewer #1: This is a generally well-written manuscript based on data collected prospectively in the UK Biobank Study. The association of daytime napping with CKD and ESKD is positive based on robust cox regression analysis despite adjustment for potential confounders. The authors also examined interaction of daytime sleepiness and sleep duration and did not find significant interaction between these two variables.

One of the major criticisms this reviewer has is that the authors do not adequately speculate the potential mechanisms in the discussion section. They do not explicitly call out the potential role of obstructive sleep apnea (OSA) as the potential mediator that explains the association between daytime sleepiness and CKD or ESKD. Daytime sleepiness and therefore napping during the day are often symptoms of underlying OSA with many negative consequences.

R: Thank you for bringing obstructive sleep apnea (OSA) to our attention. We acknowledge that our analysis and discussion did not extensively consider OSA. In the context of the relationship between daytime sleepiness and CKD/ESKD, we tend to regard OSA as a potential confounding factor rather than a mediator. This inclination arises from the notion that OSA could contribute to poor sleep quality and daytime sleepiness, consequently leading to an increase in daytime napping duration. To comprehensively evaluate the influence of OSA on the association between daytime napping and CKD/ESKD, we performed a sensitivity analysis. Given OSA's potential role as a confounder, we included it as a covariate in the Cox regression analysis. The results from this sensitivity analysis were in line with those of the primary analysis, affirming the robustness of our findings. (Rebuttal Table 1)

A limitation is the detection of CKD by ICD codes as opposed to lab values. The authors do acknowledge this weakness.

R: Thank you for your comments. we have indeed taken this limitation into consideration. We have discussed this limitation in our manuscript:

“Last, the diagnosis of CKD and ESKD mainly depends on the ICD codes, which may result in missed diagnoses, especially for CKD.” (P27)

While not absolutely necessary, the manuscript could benefit from inclusion of a DAG (causal diagram) explaining the potential pathways from exposure to outcomes and include confounders and potential mediators in such a diagram.

R: Thank you for the excellent suggestion. We have incorporated S1 Fig and Rebuttal Fig 1 as a DAG in accordance with your advice, which illustrates potential pathways from exposure to outcomes and includes confounders and potential mediators.

Rebuttal Fig 1. Directed acyclic graph illustrating the relationships between exposures, outcomes, potential mediators and potential confounders in the study.

Reviewer #2:  

MAJOR CONCERNS:

In the result session, when the cruel model (subgroup A- CKD) is adjusted for model 3 (see figure 2) the Hazard Ratio (HR) value decreases from 1.46 to 1.07 for individuals who sometimes took a nap and HR value decreases from 1.90 to 1.09 for those who usually took a nap. In both cases HR decreases in a meaningful way. The Full model’s HR values indicate a 7% (1.07) and a 9% (1.09) higher risk of CKD in individuals taking sometimes or usually a nap, respectively, when compared with the reference group (no nap). Such values are indicative of a positive association but not a consistent association as defined by the authors across the whole manuscript. (please note that a value of HR of 1.00 indicates no differences among reference group and investigated group). A meaningful decrease of the HR value is shown also for subgroup B (Figure 2) where HR diminishes from 2.07 to 1.07 for individuals who took sometimes a nap; and from 3.80 to 1.24 for those who usually took a nap. In this case, the decrease of the HR values matches an increase in P values exceeding the set limit of 0.05. Therefore the association among daily nap and ESKD looks quite weak to be claimed. 

R: Thank you for your comments. We appreciate your perspective and agree with your point. Indeed, the effect sizes approaching 1 and the p-values suggest a non-significant association between daytime napping and ESKD. We have made the necessary revisions in the discussion section to reflect this, as follows:

“In this extensive prospective study involving a middle-to-older-aged British population with an 11 years of follow-up, daytime napping was identified as a factor associated with an elevated risk of incident CKD. However, insufficient evidence indicated that daytime napping was associated with ESKD.” (P23)

Furthermore, the delta among HR values (“sometimes nap” against “usually nap”) of subgroup A and subgroup B is too small to claim “a dose-dependent fashion” for the association.

R: Thank you for pointing that out. We concur with your perspective. As you mentioned, the difference in HR values between subgroup A and subgroup B (“sometimes nap” versus “usually nap”) is negligible. We have addressed these issues in the discussion section of the manuscript. The revised paragraph is as follows:

“In this extensive prospective study involving a middle-to-older-aged British population with an 11 years of follow-up, daytime napping was identified as a factor associated with an elevated risk of incident CKD. However, insufficient evidence indicated that daytime napping was associated with ESKD.” (P23)

“In conclusion, self-reported daytime napping was positively associated with an increased incident CKD risk, suggesting that napping may be a potential risk factor for the onset of CKD.” (P28) 

The meaningful decrease of HR values in both subgroups (A and B) reported after model adjustment may suggest that major causes of CKD and ESKD are embedded in confounders. The weak positive association between day napping and CKD and ESKD is indeed in agreement with the second finding of the authors, that is, a missing addictive interaction between daytime napping and sleep duration.

R: Thank you for your feedback. I appreciate your attention to detail. I have revised the content in the discussion section as per your suggestions, as follows:

“Napping may predispose to incident CKD by increasing the risk of developing established CKD risk factors. For example, a systematic review suggested that among older adults aged >60 years, dose-response associations of daytime napping with higher odds of diabetes, dyslipidemia, metabolic syndrome and mortality were observed, starting from 0 min/d [46]. In a study during a median follow-up of 8.1 years, regular long (> 60 min) midday nap was associated with an increased hazard ratio of cardiac events [13]. In contrast, an observational prospective cohort study from Swiss population found that subjects who nap once or twice per week have a lower risk of incident CVD events, while no association was found for more frequent napping or napping duration [47].These studies additionally indicated that napping could be a modifiable factor influencing these well-established CKD risk factors. However, in our cohort, adjusting for the development of hypertension, diabetes, CVD and dyslipidemia at baseline did not materially change effect estimates for the risk of incident CKD with daytime napping. This suggests that daytime napping is associated with the risk of incident CKD via a mechanism independent of these factors.” (P24-25)

Please let me know if there are any further adjustments needed or if you have additional feedback.

Finally, day napping cannot be claimed as “an independent risk factor” cause it is not known whether day napping is induced by further confounders that have not been taken into account in the present study. The same authors are aware of that indicating that “the possibility of residual confounding by declining health cannot be ruled out”. It is therefore not known whether managing the day napping would mean a better health for the individuals.

R: Thank you for pointing that out. We appreciate your perspective, and we agree with your point. The terminology 'an independent risk factor' was indeed inappropriate, and we have made the necessary modification, as follows:

“Our study revealed a positive association between daytime napping and an elevated risk of incident CKD, indicating that it could be a potential risk factor for the onset of CKD.” (P24) 

“In conclusion, self-reported daytime napping was positively associated with an increased incident CKD risk, suggesting that napping may be a potential risk factor for the onset of CKD.” (P28)

MINOR CONCERNS:

In the “Statisitcal analyses” the authors need to explain how they calculated the “total-person time” in order to estimate the maximum confounders to be taken into account.

R: Thank you for your inquiry. The confounders used in this study were determined based on medical background knowledge, and all confounder data were collected at baseline. The 'time' used in the Cox regression was determined according to the following four scenarios:

-If a subject had no endpoint events, no loss to follow-up, and no deaths, the 'time' used for the Cox regression was the date on which we obtained the data, which was November 12, 2021; with an event record of 0.

-If a subject experienced an endpoint event, the 'time' used for the Cox regression was the date of the endpoint event occurrence, with an event record of 1.

-If a subject had no endpoint events, experienced loss to follow-up, and no deaths, the 'time' used for the Cox regression was the date of loss to follow-up, with an event record of 0.

-If a subject had no endpoint events, no loss to follow-up, but experienced death, the 'time' used for the Cox regression was the date of death, with an event record of 0.

We have attempted to address your question with the above explanation. If our response does not fully satisfy your inquiry, please let us know, and we will be happy to provide further clarification on this matter.

Major details regarding the application of the R software should be provided in the same session.

R: Thank you for pointing that out. We appreciate your identification of the issue. Concerning the application of R software in the manuscript, we have made the necessary revisions. The modified paragraph is provided below:

“All analyses were conducted utilizing R version 4.0.3. The following packages were applied: 'mice', 'dplyr', 'dplyr', 'stringr', 'survival', 'survminer', 'cmprsk', 'riskRegression', 'car', 'Publish', 'rms', 'tableone', 'mice', 'forestplot', and 'boot'.” (P12)

Finally, the reviewer also suggests to provide, as Supplementary data, the dataset applied for building multivariable COX proportional hazard regression models.

R: We express our gratitude for your comprehensive review of our manuscript. However, according to the regulations of the UK Biobank, we are unable to provide the dataset. The relevant statements are as follows: 

“The UK Biobank resource is available to bona fide researchers for health-related research in the public interest. All researchers who wish to access the research resource must register with UK Biobank by completing the registration form in the Access Management System (AMS – https://bbams.ndph.ox.ac.uk/ams/).” (P29)

---

## [Decision Letter · Decision Letter 1]

13 Dec 2023

PONE-D-23-22161R1Association of daytime napping with incidence of chronic kidney disease and end-stage kidney disease: a prospective observational studyPLOS ONE

Dear Dr. Lu,

Thank you for submitting your manuscript to PLOS ONE. After careful consideration, we feel that it has merit but does not fully meet PLOS ONE’s publication criteria as it currently stands. Therefore, we invite you to submit a revised version of the manuscript that addresses the points raised during the review process.

We look forward to receiving your revised manuscript.

Kind regards,

Giuseppe Remuzzi

Academic Editor

PLOS ONE

Journal Requirements:

Reviewers' comments:

Reviewer's Responses to Questions

**Comments to the Author**

1. If the authors have adequately addressed your comments raised in a previous round of review and you feel that this manuscript is now acceptable for publication, you may indicate that here to bypass the “Comments to the Author” section, enter your conflict of interest statement in the “Confidential to Editor” section, and submit your "Accept" recommendation.

Reviewer #2: (No Response)

Reviewer #3: All comments have been addressed

2. Is the manuscript technically sound, and do the data support the conclusions?

Reviewer #2: Yes

Reviewer #3: Yes

3. Has the statistical analysis been performed appropriately and rigorously? 

Reviewer #2: Yes

Reviewer #3: Yes

4. Have the authors made all data underlying the findings in their manuscript fully available?

Reviewer #2: Yes

Reviewer #3: Yes

5. Is the manuscript presented in an intelligible fashion and written in standard English?

Reviewer #2: Yes

Reviewer #3: Yes

6. Review Comments to the Author

Reviewer #2: The reviewer gladly realized that the suggestions have been accepted and the manuscript has been accordingly modified and improved.

The reviewer finally suggests to take care of these few points before final acceptance.

Final suggestions:

1) At page 10, in the final line the colon is repeated

2) At page 17, the sentence “In the full model, with further adjustments, although the estimates were showed a gradual attenuation, the association between napping an elevated risk of CKD remained consistent” is better rephrased:

“In the full model, with further adjustments, although the estimates showed a meaningful attenuation, the association between napping an elevated risk of CKD remained consistent”

(please note that the decrease from 1.90 to 1.09 is pretty large!)

3) At page 17, the sentence “In the full model, there was no consistent association found between napping and an elevated risk of ESKD (P for trend =.060).” is better rephrased, for example:

“In the full model, association was found between napping and elevated risk of ESKD as well. However, the P-values (0.377 and 0.060, respectively) do not allow to exclude the effect of confounders embedded in the data. Therefore, an association between napping and elevated risk of ESKD cannot be asserted”

4) At page 23, during the discussion session, it would be beneficial to add a few lines where the authors highlight their findings of an association between napping and both CKD and ESKD and that the P-values where applied as element of acceptance or rejection of the association.

5) At page 28, the statement “These findings suggested that napping was an unrecognized and clinically significant risk factor for incident CKD” is a strong conclusion that cannot be supported by the presented data. The reviewer recommends its removal.

Reviewer #3: Dear Author,

I trust this message finds you well. I have reviewed your manuscript titled "**Association of daytime napping with incidence of chronic kidney disease and end-stage kidney disease: a prospective observational study**" and found it to be intriguing and well-articulated.

In the context of your research on the association between daytime napping and chronic kidney disease (CKD), I would like to draw your attention to a recent study that might complement and strengthen your findings. The study is titled "**Association between sleep parameters and chronic kidney disease: findings from Iranian Ravansar Cohort Study**" by Hemati et al. published in BMC Nephrology (DOI: 10.1186/s12882-023-03177-3).

The Iranian Ravansar Cohort Study provides valuable insights into the relationship between sleep parameters and CKD, offering potential additional evidence to support your investigation. Considering the alignment in the research focus, incorporating findings from this Iranian study could enhance the comprehensiveness of your analysis and contribute to a more robust interpretation of the association between daytime napping and CKD.

I recommend reviewing the mentioned study and evaluating the possibility of integrating relevant insights into your work.

Best regards,

7. PLOS authors have the option to publish the peer review history of their article (what does this mean?). If published, this will include your full peer review and any attached files.

Reviewer #2: No

Reviewer #3: **Yes: **Ebrahim Norouzi

---

## [Author Response · Author response to Decision Letter 1]

8 Jan 2024

Rebuttal Letter

Dec 23, 2023 

Dear Editor,

 We sincerely thank your insightful feedback and suggestions on our manuscript titled “Association of daytime napping with incidence of chronic kidney disease and end-stage kidney disease: a prospective observational study”. We greatly appreciate this chance to further refine and enhance our work. 

In response, we have diligently addressed each comment and made the necessary revisions. All changes have been highlighted within the manuscript. Please do not hesitate to point out any further issues or concerns.

Best regards, 

Guoyuan Lu 

Journal Requirements:

R：Thank you for your reminder. we have already reviewed the reference list to ensure that it is complete and correct. At the same time, a citation has been added to the discussion section of the article, resulting in a modification to the reference list, as shown below:

“In contrast, a study from iranian ravansar cohort demonstrated that after adjusting for confounding factors, no statistically significant association of daytime napping with CKD was observed among Iranian adults in the Kurdish region [46].” (P24) 

“46. Hemati N, Shiri F, Ahmadi F, Najafi F, Moradinazar M, Norouzi E, Khazaie H.Association between sleep parameters and chronic kidney disease: findings 

from iranian ravansar cohort study. BMC Nephrol. 2023 May 17;24(1):136. 

doi: 10.1186/s12882-023-03177-3.” (P34)

Reviewer #2: The reviewer gladly realized that the suggestions have been accepted and the manuscript has been accordingly modified and improved.

The reviewer finally suggests to take care of these few points before final acceptance.

Final suggestions:

1) At page 10, in the final line the colon is repeated.

R: Thank you for pointing that out. We have already removed the unnecessary colon at the end of the last line on page 11.

2) At page 17, the sentence “In the full model, with further adjustments, although the estimates were showed a gradual attenuation, the association between napping an elevated risk of CKD remained consistent” is better rephrased: “In the full model, with further adjustments, although the estimates showed a meaningful attenuation, the association between napping an elevated risk of CKD remained consistent”

(please note that the decrease from 1.90 to 1.09 is pretty large!)

R: Thank you for your comprehensive examination of our manuscript. We have made the following modifications based on your suggestions：

“In the full model, with further adjustments, although the estimates showed a meaningful attenuation, the association between napping and elevated risk of CKD remained consistent (P for trend = .004). ” (P17)

3) At page 17, the sentence “In the full model, there was no consistent association found between napping and an elevated risk of ESKD (P for trend =.060).” is better rephrased, for example: “In the full model, association was found between napping and elevated risk of ESKD as well. However, the P-values (0.377 and 0.060, respectively) do not allow to exclude the effect of confounders embedded in the data. Therefore, an association between napping and elevated risk of ESKD cannot be asserted”

R: Thank you for your thorough review of our manuscript. Your expression is very reasonable. We have already made modifications in accordance with your suggestions, as follows:

“In the full model, after adjusting for covariates in model 3, relative to the reference group, the HRs (95% CIs) for ESKD were 1.07 (0.92 to 1.24) in the group that occasionally naps and 1.24 (0.99 to 1.54) in the group that usually naps (as illustrated in Fig 2). A correlation was observed between napping and an increased risk of ESKD as well. Nevertheless, the P-values (0.377 and 0.060, respectively) did not allow to exclude the effect of confounding factors embedded in the data. Therefore, an association between napping and elevated risk of ESKD could not be asserted.” (P17-18)

“After adjusting for these factors, no statistically significant association of daytime napping with ESKD was observed, indicating that these factors might confound the observed associations between daytime napping and ESKD.” (P27) 

4) At page 23, during the discussion session, it would be beneficial to add a few lines where the authors highlight their findings of an association between napping and both CKD and ESKD and that the P-values where applied as element of acceptance or rejection of the association.

R: Thank you for your feedback. We appreciate your attention to detail. We have revised the content in the discussion section as per your suggestions, as follows:

“Our study revealed a positive association between daytime napping and an elevated risk of incident CKD. After adjusting for confounding factors, although their relationship weakened, the p-value for trend still hold statistical significance. The variations in these findings might stem from differences among participants in terms of race, lifestyle, and health. Consequently, future studies should take into account the varying risk levels for CKD across diverse racial and ethnic groups.” (P24)

“After adjusting for confounding factors, the P-value for trend (= 0.060) was no longer statistically significant. Therefore, it could not be conclusively stated that a relationship exists between napping and an increased risk of ESKD.” (P26) 

5) At page 28, the statement “These findings suggested that napping was an unrecognized and clinically significant risk factor for incident CKD” is a strong conclusion that cannot be supported by the presented data. The reviewer recommends its removal.

R: Thank you for the excellent suggestion. We appreciate your perspective and agree with your point. We have made the necessary revisions in the conclusion section to reflect this, as follows:

“In conclusion, self-reported daytime napping was positively associated with an increased incident CKD risk, suggesting that napping may be a potential risk factor for the onset of CKD. However, the specific mechanism by which daytime napping induces the occurrence of CKD is not well understood; Future work is needed to explore the specific mechanisms and assess whether the observed relationship between napping with incident CKD is causal.” (P28)

Reviewer #3: 

I trust this message finds you well. I have reviewed your manuscript titled "**Association of daytime napping with incidence of chronic kidney disease and end-stage kidney disease: a prospective observational study**" and found it to be intriguing and well-articulated.

R：Thank you very much for your affirmation and recognition of the article titled “Association of daytime napping with incidence of chronic kidney disease and end-stage kidney disease: a prospective observational study.” Your affirmation and recognition have further boosted our confidence. We will implement further edits to the article as per your suggestions to enhance its overall clarity and expression.

In the context of your research on the association between daytime napping and chronic kidney disease (CKD), I would like to draw your attention to a recent study that might complement and strengthen your findings. The study is titled "**Association between sleep parameters and chronic kidney disease: findings from Iranian Ravansar Cohort Study**" by Hemati et al. published in BMC Nephrology (DOI: 10.1186/s12882-023-03177-3).

The Iranian Ravansar Cohort Study provides valuable insights into the relationship between sleep parameters and CKD, offering potential additional evidence to support your investigation. Considering the alignment in the research focus, incorporating findings from this Iranian study could enhance the comprehensiveness of your analysis and contribute to a more robust interpretation of the association between daytime napping and CKD.

I recommend reviewing the mentioned study and evaluating the possibility of integrating relevant insights into your work.

R：Thank you for providing us with detailed guidance and advice. We have carefully read and understood the content of the article titled "**Association between sleep parameters and chronic kidney disease: findings from Iranian Ravansar Cohort Study**" based on your suggestions. Accordingly, we have made relevant modifications in line with the results and discussions of the article. The revisions provide a stronger explanation of the relationship between daytime napping and CKD. The modified content is as follows:

“In contrast, a study from iranian ravansar cohort demonstrated that after adjusting for confounding factors, no statistically significant association of daytime with CKD was observed among Iranian adults in the Kurdish region [46]. Even after accounting for potential confounding factors, our study still indicated an association between daytime napping and an increased risk of developing CKD. Despite a weakened association, the p-value for the trend (= .040) remains statistically significant. The variations in these findings might stem from differences among participants in terms of race, lifestyle, and health. Consequently, future studies should take into account the varying risk levels for CKD across diverse racial and ethnic groups.” (P24)

---

## [Decision Letter · Decision Letter 2]

24 Jan 2024

Association of daytime napping with incidence of chronic kidney disease and end-stage kidney disease: a prospective observational study

PONE-D-23-22161R2

Dear Dr. Lu,

We’re pleased to inform you that your manuscript has been judged scientifically suitable for publication and will be formally accepted for publication once it meets all outstanding technical requirements.

Kind regards,

Giuseppe Remuzzi

Academic Editor

PLOS ONE

Additional Editor Comments (optional):

Reviewers' comments:

Reviewer's Responses to Questions

**Comments to the Author**

1. If the authors have adequately addressed your comments raised in a previous round of review and you feel that this manuscript is now acceptable for publication, you may indicate that here to bypass the “Comments to the Author” section, enter your conflict of interest statement in the “Confidential to Editor” section, and submit your "Accept" recommendation.

Reviewer #2: All comments have been addressed

Reviewer #3: All comments have been addressed

2. Is the manuscript technically sound, and do the data support the conclusions?

Reviewer #2: Yes

Reviewer #3: Yes

3. Has the statistical analysis been performed appropriately and rigorously? 

Reviewer #2: Yes

Reviewer #3: Yes

4. Have the authors made all data underlying the findings in their manuscript fully available?

Reviewer #2: Yes

Reviewer #3: Yes

5. Is the manuscript presented in an intelligible fashion and written in standard English?

Reviewer #2: Yes

Reviewer #3: Yes

6. Review Comments to the Author

Reviewer #2: In response to my queries that authors have adequately responded. The paper can be accepted and published

Reviewer #3: (No Response)

7. PLOS authors have the option to publish the peer review history of their article (what does this mean?). If published, this will include your full peer review and any attached files.

Reviewer #2: No

Reviewer #3: **Yes: **Ebrahim Norouzi

---

## [Editor Report · Acceptance letter]

12 Mar 2024

PONE-D-23-22161R2 

PLOS ONE

Dear Dr. Lu, 

I'm pleased to inform you that your manuscript has been deemed suitable for publication in PLOS ONE. Congratulations! Your manuscript is now being handed over to our production team.

Kind regards, 

on behalf of

Prof. Giuseppe Remuzzi 

Academic Editor

PLOS ONE